# Decreasing pdzd8-mediated mito–ER contacts improves organismal fitness and mitigates A$\beta_{42}$ toxicity

Victoria L Hewitt[1,2], Leonor Miller-Fleming[1], Madeleine J Twyning[1], Simonetta Andreazza[1], Francesca Mattedi[3], Julien Prudent[1], Franck Polleux[2,4,5], Alessio Vagnoni[3], Alexander J Whitworth[1]

Mitochondria-ER contact sites (MERCs) orchestrate many important cellular functions including regulating mitochondrial quality control through mitophagy and mediating mitochondrial calcium uptake. Here, we identify and functionally characterize the Drosophila ortholog of the recently identified mammalian MERC protein, Pdzd8. We find that reducing pdzd8-mediated MERCs in neurons slows age-associated decline in locomotor activity and increases lifespan in Drosophila. The protective effects of pdzd8 knockdown in neurons correlate with an increase in mitophagy, suggesting that increased mitochondrial turnover may support healthy aging of neurons. In contrast, increasing MERCs by expressing a constitutive, synthetic ER–mitochondria tether disrupts mitochondrial transport and synapse formation, accelerates age-related decline in locomotion, and reduces lifespan. Although depletion of pdzd8 prolongs the survival of flies fed with mitochondrial toxins, it is also sufficient to rescue locomotor defects of a fly model of Alzheimer's disease expressing Amyloid $\beta_{42}$ (A$\beta_{42}$). Together, our results provide the first in vivo evidence that MERCs mediated by the tethering protein pdzd8 play a critical role in the regulation of mitochondrial quality control and neuronal homeostasis.

## Introduction

Because the vast majority of neurons are postmitotic, maintaining functional neurons throughout an organism's lifetime requires tight regulation of organelle functions and cellular homeostatic mechanisms. Mitochondria and the ER extend throughout neuronal compartments including axons and dendrites, and both are vital and interdependent contributors to neuronal health (Wu et al, 2017). Mitochondria-ER contacts (MERCs) are controlled by a variety of contact site-resident proteins and contribute to a range of functions required for proper development and maintenance of postmitotic neurons, including regulation of calcium homeostasis, lipid biogenesis, organelle reshaping and dynamics, and metabolic signalling (Giacomello & Pellegrini, 2016; Paillusson et al, 2016). Dysregulation of MERCs is particularly damaging to neurons as they are especially susceptible to calcium overload, oxidative and ER stresses, and to altered mitochondrial function, localization, and transport (Misgeld & Schwarz, 2017; Lee et al, 2018a).

As MERCs are modulated by a number of different protein complexes, the detrimental effects of MERCs dysregulation are varied because of the diversity of these contact site functions (Martino Adami et al, 2019). The critical function of MERCs in regulating cellular responses to damage and stress is underscored by the finding that many human patient cellular models and animal models of age-related neurodegenerative diseases have been shown to have disrupted MERCs. Both reduced MERCs (Sepulveda-Falla et al, 2014; De Vos & Hafezparast, 2017) and increased MERCs (Zampese et al, 2011; Area-Gomez et al, 2012; Gómez-Suaga et al, 2019) have been implicated in neurodegenerative diseases. Consequently, there is still little consensus on how altered MERCs contribute to neurodegeneration, even within a single disease model (Erpapazoglou et al, 2017). The various functions of MERCs make it likely that multiple mechanisms might be involved and, with an ever-expanding toolkit, we can now better define the molecular identities and specific functions of ER–mitochondria tethering complexes and begin to unify many of the seemingly conflicting discoveries in this rapidly growing field (Csordas et al, 2018).

Pdzd8 is one of the most recently identified proteins that mediates mammalian MERCs (Hirabayashi et al, 2017) and is proposed to be a paralog of Mmm1 (Wideman et al, 2018), a component of the fungal-specific ER mitochondria encounter structure and first MERC complex identified (Kornmann et al, 2009). In *Drosophila melanogaster* pdzd8 is highly expressed in neurons and, therefore, we explored the consequences of its depletion by RNAi in neurons both at the cellular and at the organismal level. Importantly, we also describe how the phenotypes associated with neuron-specific depletion of pdzd8 change with age and may contribute to healthy aging. We show the MERCs mediated by pdzd8 tethering regulate

[1]Medical Research Council, Mitochondrial Biology Unit, University of Cambridge, Cambridge, UK   [2]Department of Neuroscience, Columbia University Medical Center, New York, NY, USA   [3]Department of Basic and Clinical Neuroscience, Maurice Wohl Clinical Neuroscience Institute, IoPPN, King's College London, London, UK   [4]Mortimer B Zuckerman Mind Brain Behavior Institute, New York, NY, USA   [5]Kavli Institute for Brain Sciences, Columbia University Medical Center, New York, NY, USA

Correspondence: a.whitworth@mrc-mbu.cam.ac.uk

mitochondrial turnover through mitophagy and that reducing these contacts prolonged locomotor activity and lifespan in *Drosophila*. We also report that reducing excess MERCs observed in an Aβ42 model of Alzheimer's disease (AD) correlates with improved motor performance in this model. Together, our results show the critical contribution of MERCs to neuronal and organismal homeostasis by elucidating the consequence of excess and reduced MERCs in an in vivo model.

## Results

### Characterization of fly pdzd8

The *D. melanogaster* gene *CG10362* encodes an uncharacterized protein in the PDZK8 family (Lee & Hong, 2006). The product of *CG10362* has a similar predicted domain structure to mammalian Pdzd8 (Fig 1A). Expression of *CG10362* in flies is low but is most highly expressed in the central nervous system (FlyAtlas 2; Fig S1A) (Leader et al, 2018) and is enriched in neurons over glia (Fig S1B) (Davie et al, 2018). This specificity in expression in *Drosophila* provided not only an excellent opportunity to explore the neuronal functions of this newly discovered MERCs protein but also to investigate the functional relevance of MERCs in adult and aging neurons. Based on the sequence homology and functional analysis presented here, we propose that *CG10362* encodes the fly ortholog of mammalian *Pdzd8* and will hereafter refer to *CG10362* as *pdzd8*.

To characterize the function of pdzd8 in flies, we used the GAL4–UAS system to manipulate its expression (Brand & Perrimon, 1993). Ubiquitous expression of an RNAi construct targeting *pdzd8* strongly reduces its mRNA levels in larvae (Fig S1C). To determine whether MERCs were decreased in neurons expressing *pdzd8*-RNAi, we analysed adult fly brains by transmission electron microscopy (TEM) and manually identified contacts between ER and mitochondria, with the experimenter blind to conditions (Figs 1B and S1D). These results showed that the proportion of mitochondria in contact with ER in the soma of adult fly neurons is significantly reduced in flies expressing *pdzd8*-RNAi compared with controls (Fig 1C).

To confirm that *pdzd8*-RNAi reduces MERCs in axons as well as soma, we adapted the recently developed MERC quantification tool, SPLICS, a split-GFP-based contact site sensor (Cieri et al, 2017), to create a SPLICS transgenic reporter line. The SPLICS construct targets β-strands 1–10 of GFP to the mitochondrial outer membrane and β-strand 11 to the ER membrane. Where these membranes are in close proximity, fluorescent puncta are produced by reconstitution of the split-GFP (Fig S2A). To validate this tool in *Drosophila*, we expressed SPLICS in motor neurons (Fig S2B) and compared the number of puncta in control axons and those expressing a well characterized artificial mitochondria-ER tether developed by Csordas et al (2006) and Basso et al (2018). Expression of this synthetic mitochondria-ER tether induces formation of ~5 nm MERCs through targeting motifs that anchor it in both the mitochondrial outer membrane and the ER (Fig S2C). The density of SPLICS puncta in the axons expressing the tether was four times higher than controls (Fig S2D and E), indicating that the SPLICS

reporter was able to report a change in MERCs resulting from expression of the synthetic tether in neurons in vivo.

Using this SPLICS construct, we detected a significant decrease in the density of SPLICS puncta in central larval axons (bundles projecting to segments A7 and A8) expressing the *pdzd8*-RNAi compared with *LacZ*-RNAi controls (Fig 1D and E). Whereas *pdzd8* expression is low beyond the nervous system (Fig S1A), ubiquitous knockdown of *pdzd8* also reduced the extent of MERCs measured by fluorescence colocalization of ER and mitochondrial signals by super-resolution microscopy (structured illumination microscopy) in larval epidermal cells (Fig 1F–H), corroborating our results using TEM and SPLICS analysis.

Consistent with the first reported function of Pdzd8 at MERCs in mouse neurons (Hirabayashi et al, 2017), we observed a reduced number of MERCs (Fig 1), but no obvious changes in mitochondrial or ER morphology in fly larval or adult neurons upon *pdzd8* knockdown (See Figs 1B, D, and F, S1D and E, 4A and B and F–I, and 5A and C). Together, these data support that *CG10362* encodes the *Drosophila* ortholog of Pdzd8 and functions like mammalian Pdzd8 to regulate MERCs. We next sought to investigate the consequences of loss of pdzd8 from neurons on organismal phenotypes.

### Reduced mitochondria-ER contacts are protective in aging neurons

Knockdown of *pdzd8* in fly neurons produces viable adults. There was no impact on the locomotor performance of pan-neuronal *pdzd8*-RNAi knockdown in young flies assessed in a climbing assay (Fig 2A). Surprisingly, we found that loss of pdzd8 dramatically slowed the age-associated decline in climbing ability (Fig 2A). To corroborate these striking results, we repeated the analysis alongside additional independent controls including another control RNAi (*luciferase*-RNAi) and another inert transgene (*mitoGFP*), with equivalent results (Fig S3). Importantly, this effect was also reproduced by motor neuron-specific *pdzd8* knockdown (Fig S4A). Strikingly, the increase in locomotor activity was accompanied by a significant increase in lifespan (Fig 2C).

In contrast, increasing MERCs by expression of a synthetic mitochondria-ER tether in all neurons resulted in a climbing defect in young flies and a significant acceleration of the age-related decline in climbing (Fig 2B), consistent with previous reports (Basso et al, 2018). This climbing defect was exacerbated with age (Fig 2B) and associated with a substantially reduced lifespan (Fig 2D). Consistent with these results, increased expression of *pdzd8* also resulted in decreased climbing ability with age (Fig S4B). Notably, this effect was suppressed by co-expression of the *pdzd8*-RNAi, further validating the specificity of this transgene (Fig S4B). Therefore, decreasing pdzd8-mediated MERCs in neurons prolonged lifespan and protected against locomotor decline with age, whereas increased MERCs in neurons accelerate the age-related decrease in locomotor activity and decreased lifespan.

### Loss of neuronal pdzd8 promotes survival in the presence of mitochondrial toxins

To investigate how reduction of pdzd8-mediated MERCs improved fitness, that is, prevented age-related decline in locomotor activity

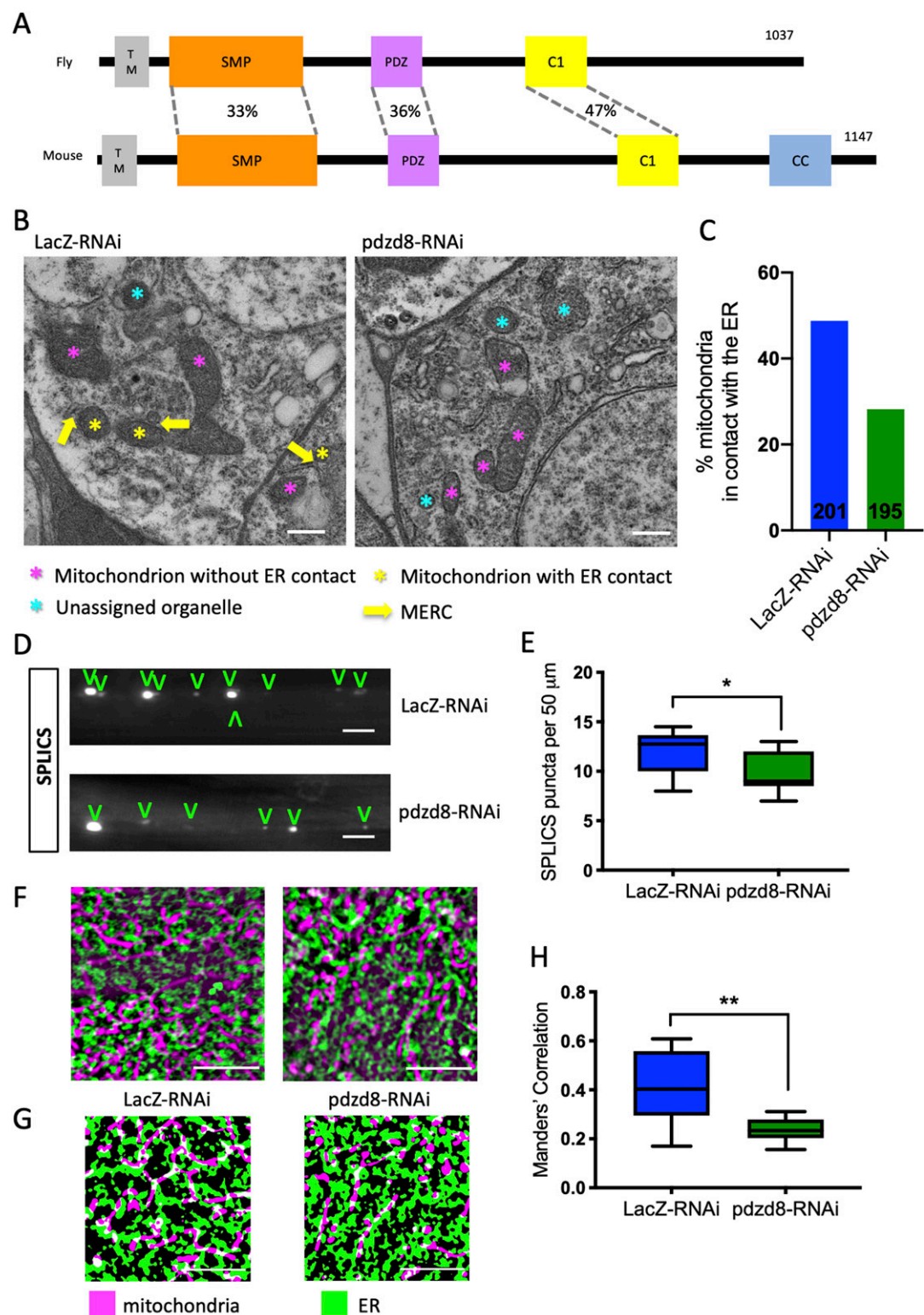

**Figure 1. Expression of *pdzd8*-RNAi reduces mitochondria-ER contacts.**
**(A)** Domain organisation of *Drosophila* pdzd8 (CG10362) compared with mouse Pdzd8 showing percentage identities of conserved domains based on Clustal Omega alignments. Overall percentage identity of the amino acid sequences is 21%. SMP (synaptotagmin-like mitochondrial lipid-binding proteins) 33% identical, PDZ (PSD95/DLG/ZO-1) 36% identical, C1 (C1 protein kinase C conserved region 1 also known as Zn finger phorbol-ester/DAG-type signature) 47% identical; TM: predicted transmembrane domain, CC: coil–coil domain. **(B)** Electron microscopy images of cell bodies of the posterior protocerebrum of 2-d-old adult brains showing representative images of ER, mitochondria, and MERCs in soma from nSyb>*LacZ*-RNAi and nSyb>*pdzd8*-RNAi flies. Scale bar 500 nm. Mitochondria without identifiable ER

and increased lifespan, we assessed whether reducing *pdzd8* expression in neurons may protect from additional stresses during aging. We assessed lifespan in flies aged on food with limited nutrients (5% sucrose, 1% agar) and found that in contrast to flies aged on a rich diet (Fig 2C), neuronal expression of *pdzd8*-RNAi no longer extended the lifespan in comparison to controls (Fig 3A). When an additional oxidative stress was introduced by adding hydrogen peroxide to the food, the flies expressing *pdzd8*-RNAi died faster than controls (Fig 3B). Thus, in the presence of general stresses, *pdzd8*-RNAi was not protective.

Because of the function of pdzd8 at MERCs, we examined whether the protective effects caused by the pdzd8 depletion in neurons were more specifically associated with mitochondrial dysfunction. To address this, we fed the flies mitochondrial toxins—rotenone, a complex I inhibitor, or antimycin A, a complex III inhibitor—to block the electron transport chain and cause dysfunctional mitochondria. Reducing pdzd8 levels in neurons significantly prolonged the survival of flies fed with both mitochondrial toxins, rotenone (Fig 3C), or antimycin A (Fig 3D), compared with control flies. As improved mitochondrial function could also contribute to the protective effects of loss of pdzd8 in neurons we measured ATP levels in young or aged fly heads expressing *pdzd8*-RNAi but found no significant differences (Fig S4C). These results indicated that neuronal loss of pdzd8 protects flies from damage specifically induced by mitochondrial toxins but not general cellular stresses.

### Modulating MERCs causes axonal transport and NMJ defects

Although mitochondrial motility is important for neuronal health, it remains an open question whether decline of mitochondrial transport in neurons contributes to aging (reviewed in Mattedi and Vagnoni [2019]). We first tested the hypothesis that decreased ER–mitochondrial tethering contributes to the protective effect of *pdzd8* down-regulation in aging through changes in mitochondrial motility. Examining the distribution and morphology of mitochondria in axons of larval CCAP neurons (Park et al, 2003), we found no significant change in mitochondrial length or density in axons when comparing control to *pdzd8*-RNAi or synthetic ER–mitochondria tether-expressing neurons (Fig 4A–C). However, while increasing tethering dramatically decreased mitochondrial motility (Fig 4D and E), loss of MERCs mediated by knockdown of *pdzd8* had no impact on axonal transport in larvae (Fig 4D and E).

To better understand the effects of altered MERCs in neurons, we analysed the morphology of mitochondria located in boutons of larval neuromuscular junctions (NMJs) on muscle 4. Knockdown of *pdzd8* led to smaller NMJs and a significant reduction of mitochondrial volume, but overall, no change in mitochondrial density compared to control flies (Fig 4F–I), showing that mitochondria distribute normally in these smaller NMJs. Increased tethering,

however, resulted in severely deformed NMJs (Fig S4D), and made type 1s and 1b synaptic boutons indistinguishable. Together, these results show that increasing tethering has dramatic and detrimental effects early in development but reduced tethering through *pdzd8*-RNAi expression has more limited effects during these early stages of neuronal development.

### Reducing *pdzd8* expression increases mitophagy in aged neurons

We hypothesized that the reduced sensitivity to mitochondrial toxins observed in *pdzd8*-RNAi flies may be due to improved mitochondrial quality control mechanisms. Thus, we analysed the levels of mitophagy, the clearance of damaged mitochondria by autophagy, in these neurons using the mitoQC mitophagy reporter (Allen et al, 2013; Lee et al, 2018b). MitoQC is a pH-sensitive mCherry-GFP fusion protein targeted to the mitochondrial outer membrane which provides a read out of mitophagy when mitochondria are targeted to the acidic lysosome where GFP is quenched leaving mCherry-only puncta. Mitophagy was detected in neuronal soma of both larval and adult brains (Fig 5A and C). Although there was no significant difference in mitophagy levels in larval or young adult neurons expressing *pdzd8*-RNAi, mitophagy was significantly increased in the brains of 20-d-old *pdzd8*-RNAi animals compared to controls (Fig 5A–D).

The majority of mitochondria in neurons are found in the neurites, so to examine mitophagy in axons of aged flies, we analysed mitoQC signal in axons of the adult fly wing in situ (Vagnoni & Bullock, 2016) (Fig 5E). Here, in contrast to the adult brain cell bodies, we observed an age-dependent increase in mitophagy in axons of control flies (Fig 5F). Furthermore, consistent with our previous results, *pdzd8* knockdown further increased mitophagy in axons of aged flies (Fig 5E and F). Together, these results indicate that loss of pdzd8 promotes the turnover of mitochondria in aging neurons, which could lead to the decreased age-associated decline in locomotor activity and increased lifespan observed in *pdzd8*-RNAi animals.

### Reduced MERCs is protective in a fly model of AD

So far, our results indicate that the loss of pdzd8 is protective against mitochondrial insults, prevents the age-related decrease in locomotion and increases lifespan. Because mitochondrial dysfunction is a common feature of many neurodegenerative diseases, and altered MERCs have been documented in some, we next sought to explore the neuroprotective potential of pdzd8 depletion in an age-related neurodegenerative disease model. To this end, we turned to an AD fly model where the expression of pathogenic $A\beta_{42}$ has been shown to cause neural dysfunction, due in part to oxidative stress (Rival et al, 2009).

---

contacts marked with magenta *, mitochondria forming ER contact marked with yellow * with yellow arrow indicating contact location, organelles that did not contain clear cristae are marked with a cyan * and were excluded from the analysis. **(C)** Percentage of mitochondria in contact with the ER from controls and pan-neuronal nSyb>*pdzd8*-RNAi flies quantified from EM images of 2-d-old adult brains. n = 3 brains per genotype, numbers on bars indicate number of mitochondria analysed. **(D)** SPLICS puncta indicating MERCs in axon bundles of larval neurons from controls and nSyb>*pdzd8*-RNAi flies. Quantified puncta highlighted with V. Scale bar 5 $\mu$m. **(D, E)** Quantification of SPLICS puncta in (D). n = 11 animals per genotype, *P* = 0.0198, unpaired *t* test with Welch's Correction. **(F)** Representative structured illumination microscopy images of ER (green, ER-Tomato) and mitochondria (purple, mitoGFP) in larval epidermal cells from controls and da>*pdzd8*-RNAi flies. Scale bar = 5 $\mu$m. **(F, G)** Binarized images of ER and mitochondria shown in (F). **(H)** Quantification of colocalization of ER and mitochondria using Mander's Correlations in 12 control cells and 14da > *pdzd8*-RNAi cells (1 field of view per cell) compared using an unpaired *t* test with Welch's Correction. *P* = 0.012.

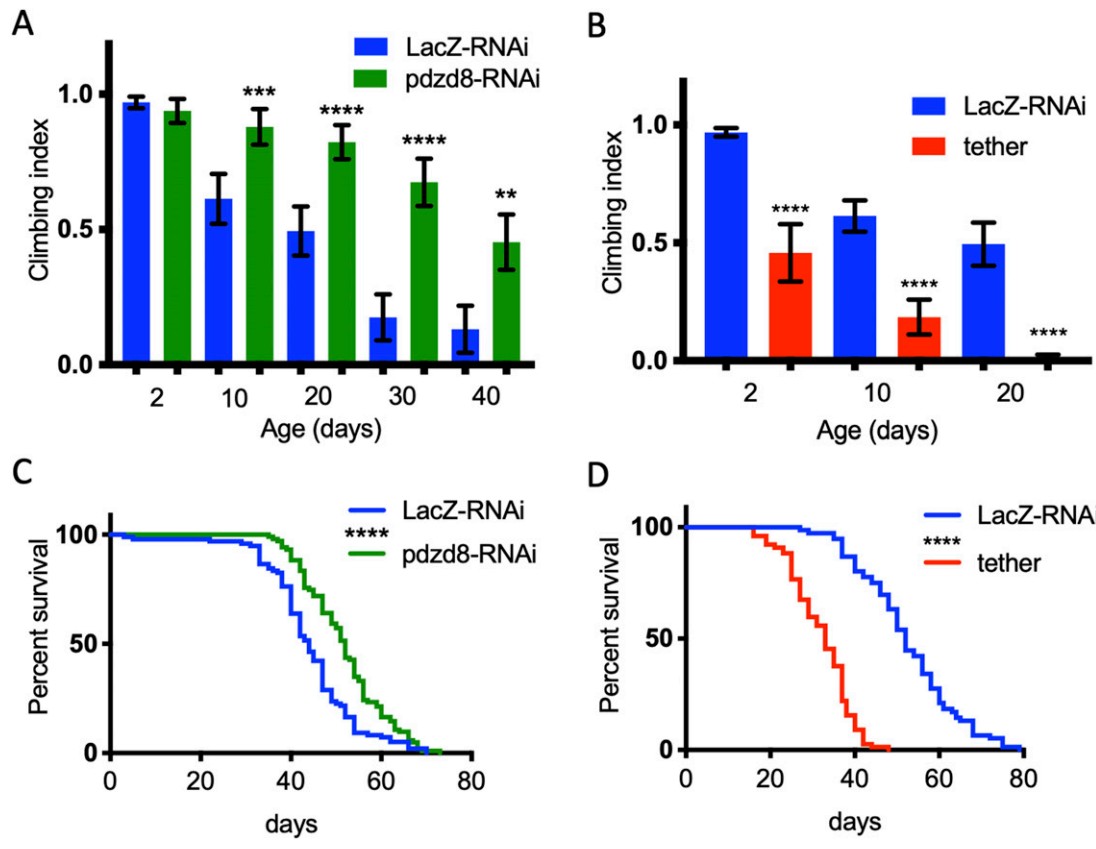

**Figure 2.  Lifespan and locomotor activity changes in aged flies with pan-neuronal driven alterations in tethering.**
**(A, B)** Locomotor activity of flies was assessed during aging by negative geotaxis climbing assays on the indicated days. n > 50 flies per genotype. Flies expressing (A) *pdzd8*-RNAi or (B) synthetic tether were compared with *LacZ*-RNAi controls. Statistical analysis was performed using Kruskal–Wallis test with Dunn's post hoc correction. **P < 0.01, ***P < 0.001, ****P < 0.0001. **(C, D)** Lifespans in standard growth conditions and food. **(C)** Flies expressing *pdzd8*-RNAi were compared with *LacZ*-RNAi controls. n = 97, 108 per genotype, median survival 44 versus 52 d, P < 0.0001. **(D)** Flies expressing the synthetic tether were compared with *LacZ*-RNAi controls. n = 74, 85 per genotype, median survival 52 versus 33 d, P < 0.0001.

Because increased MERCs have also been associated with AD (Area-Gomez et al, 2012; Del Prete et al, 2017), we first used the SPLICS reporter to determine the number of MERCs in larval axons. Consistent with other cellular and organismal models, flies expressing the A$\beta_{42}$ showed an increase in SPLICS puncta, indicating MERCs are increased in axons of this model of AD (Fig 6A and B). This was accompanied by a significant climbing defect in young flies that worsens rapidly with age (Fig 6C) (Crowther et al, 2005). Thus, we hypothesized that reducing pdzd8-mediated MERCs could be protective in this progressive neurodegenerative disease model associated with increased MERCs. Interestingly, *pdzd8* knockdown reduced MERCs back to control levels in A$\beta_{42}$ flies (Fig 6B). Moreover, reduction in *pdzd8* was sufficient to significantly restore locomotor function in young and 10-d-old A$\beta_{42}$ flies (Fig 6C).

To address the underlying mechanism behind this rescue we first hypothesized that increased mitophagy may improve neuronal function. Examining mitophagy in wing axons, we confirmed the age-dependent increase in mitophagy in control animals, which was exacerbated by A$\beta_{42}$ expression, but we found this was not further increased by *pdzd8*-RNAi (Fig 6D).

As MERCs mediate ER-to-mitochondria calcium transfer, and loss of Pdzd8 increases intracellular calcium dynamics in mouse neurons (O'Hare et al, 2022), we next explored whether calcium handling was altered in this AD model. Functional mitochondria can take up substantial amounts of calcium released from the ER before exceeding their so-called calcium retention capacity (CRC). In comparison to control flies, A$\beta_{42}$ expressing flies have a dramatically reduced CRC (Fig 6E–H). However, *pdzd8* knockdown was able to partially rescue CRC in A$\beta_{42}$ flies (Fig 6E–H). Thus, taken together, these results show that reducing pdzd8-mediated MERCs is protective in this model of AD which manifests increased MERCs. The beneficial effects are likely mediated at least in part by reduced ER–mitochondrial calcium transfer.

## Discussion

Here we have identified and characterized the putative *Drosophila* ortholog of the mammalian MERC tethering protein Pdzd8 (Hirabayashi et al, 2017). The sequence divergence between Pdzd8 and its yeast paralog Mmm1 and the additional domains present in Pdzd8 (Wong & Levine, 2017), made the relationship between these paralogs difficult to identify (Wideman et al, 2018). Whereas the

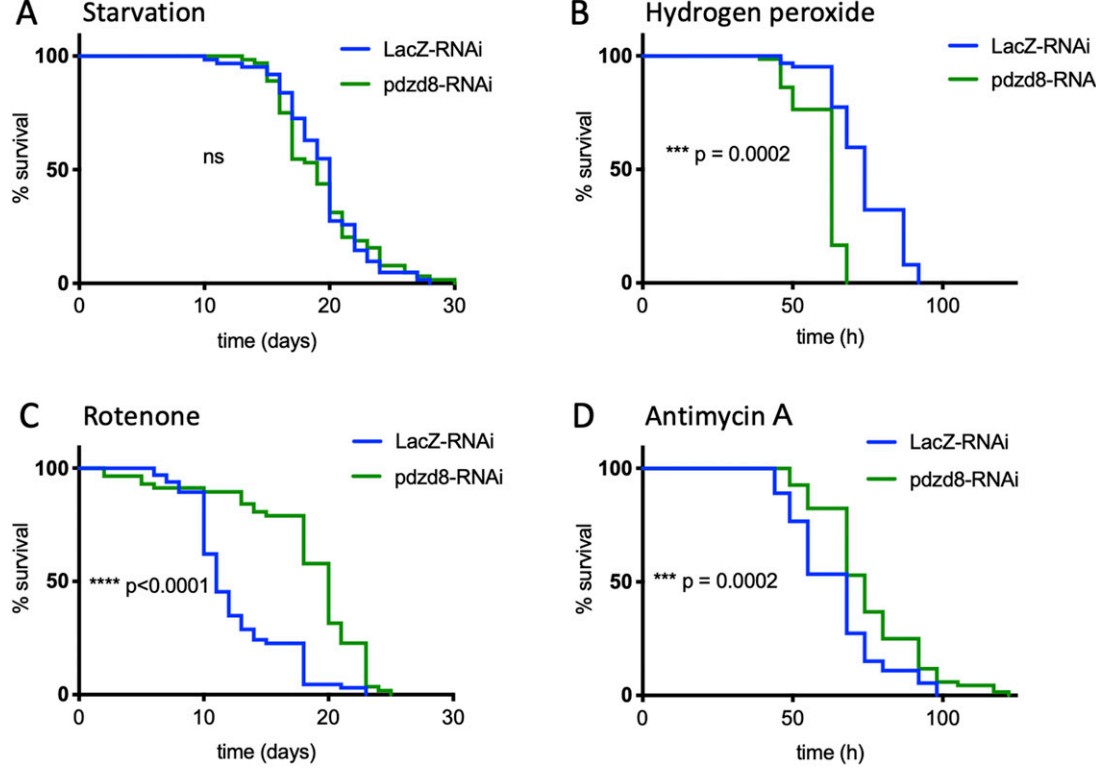

**Figure 3. Knockdown of *pdzd8* protects flies against mitochondrial toxins.**
Lifespans of flies expressing pan-neuronal *pdzd8*-RNAi were compared with *LacZ*-RNAi controls when aged on a restricted diet of food containing 1% agar with 5% sucrose. **(A)** Lifespan with dietary restriction alone. N = 62 versus 64, median survival 20 versus 19 d, difference ns. **(B)** Lifespan with addition of 5% hydrogen peroxide. Median survival: 63 versus 74 h, n = 67, 74, P = 0.0002. **(C)** Lifespan with addition of 1 mM rotenone. Median survival: 11 versus 20 d, n = 66, 57, P < 0.0001. **(D)** Lifespan with addition of 5 μg/ml antimycin A. Median survival: 74 versus 68 h, n = 72, 68, P = 0.0002.

conserved predicted domain structure strongly suggests *Drosophila CG10362* encodes the fly ortholog of mouse Pdzd8, the overall sequence identity of these proteins is low (21%), implicating more species-specific functional specialisations.

Using RNAi, we characterized the effects of pdzd8 depletion in *Drosophila* with a focus on neurons where this protein is most highly expressed. We found that knockdown of *pdzd8* reduces contacts between the ER and mitochondria in epidermal cells measured using super-resolution microscopy analysis of ER and mitochondria labeled with fluorescent reporters, in motor neurons monitored by the fluorescent contact site reporter SPLICS, and in the soma of adult neurons using TEM. These data suggest that pdzd8, like its mammalian ortholog, functions as a tether between ER and mitochondria. The only other neuronally expressed tethering protein that has been characterized in flies is the *Drosophila* ortholog of Mfn2, Marf (Hwa et al, 2002). However, the analysis of Marf is complicated by its additional roles in mitochondrial and ER morphology (Debattisti et al, 2014; Sandoval et al, 2014; El Fissi et al, 2018). Whereas mammalian Pdzd8 is expressed in a range of tissues (Hirabayashi et al, 2017), in flies, *pdzd8* mRNA expression is low outside the nervous system. Knockdown of *pdzd8* and expression of a synthetic tether therefore provided a unique opportunity to simply and selectively examine the function of MERCs in *Drosophila* neurons.

We found that increased ER–mitochondrial tethering in neurons strongly impairs climbing ability and reduces lifespan of flies, consistent with previous reports using this construct (Basso et al, 2018).

Other similar manipulations have also been shown to result in dopaminergic neuron loss (Lee et al, 2018c) and detrimental effects on sleep in ventral lateral neurons (Valadas et al, 2018). Here, we also observe highly abnormal NMJ development associated with increased MERCs, and smaller but otherwise structurally intact NMJs upon *pdzd8* knockdown. The function of the yeast paralog Mmm1 and of its Synaptotagmin-like mitochondrial lipid-binding protein (SMP) domains suggests that disruptions in lipid transfer at the mitochondria-ER interface due to less pdzd8 might contribute to these developmental defects (Jeong et al, 2017; Kawano et al, 2018; Shirane et al, 2020); however, whether lipid biogenesis defects could contribute to a protective effect of *pdzd8*-RNAi in neurons remains an open question.

Transport of mitochondria is intimately linked to the health of neurons, at least in the peripheral nervous system (De Vos et al, 2008; Harbauer, 2017; Misgeld & Schwarz, 2017). Although there is some evidence that increased MERCs may be directly associated with decreased mitochondrial motility (Krols et al, 2018), this has not been shown in neurons which are particularly sensitive to mitochondrial transport imbalance (Maday et al, 2014). Our data suggest that axonal mitochondrial transport defect contributes to the detrimental effects of increased tethering and adds to the evidence that efficient mitochondrial transport is essential for healthy aging neurons, as seen in many models of neurodegenerative motor disorders (Baldwin et al, 2016).

In contrast to the detrimental effects of increased tethering, reducing MERCs by knockdown of *pdzd8* in *Drosophila* neurons

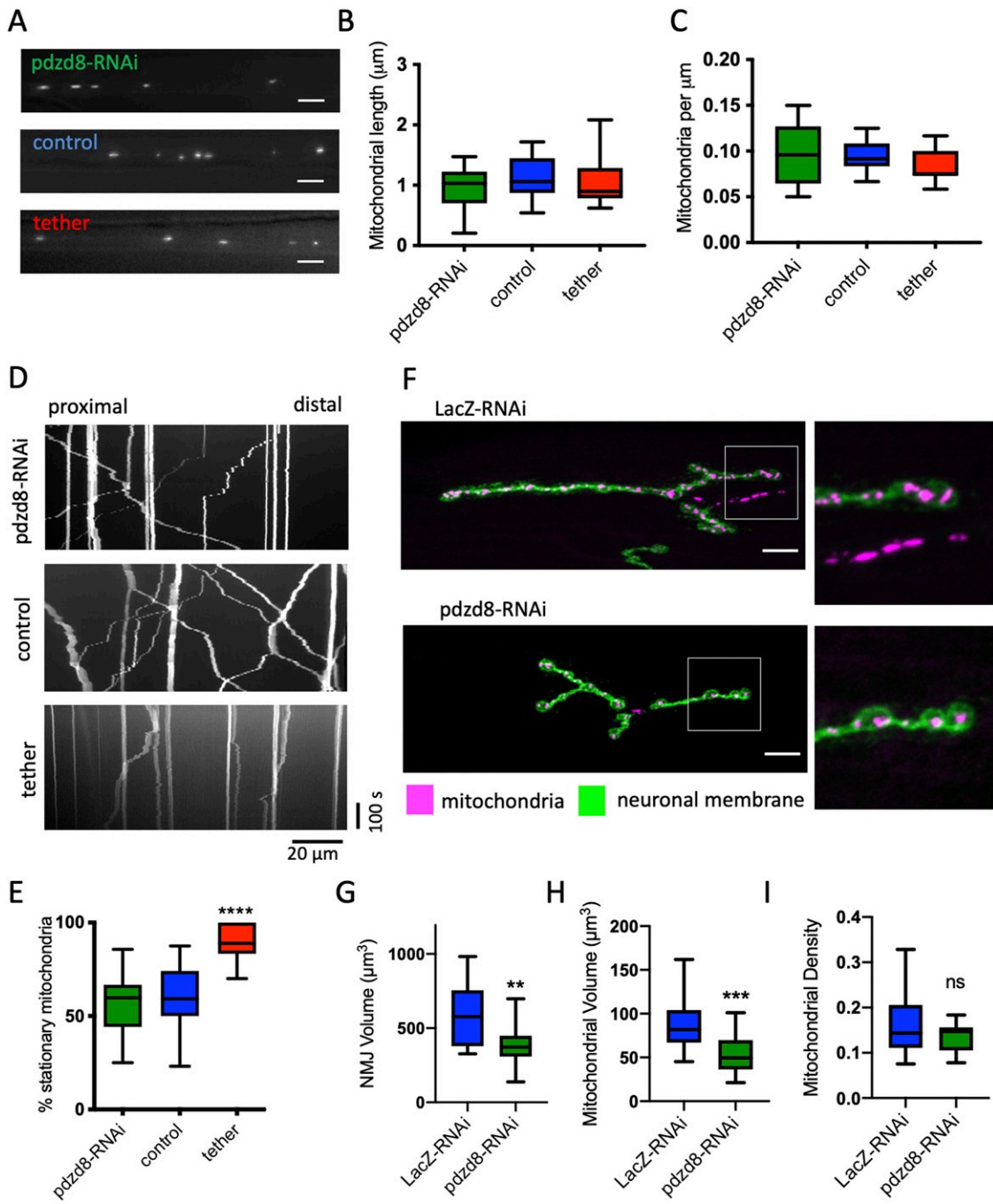

**Figure 4. Knockdown of *pdzd8* in larval neurons causes minor defects, whereas increasing MERCs is detrimental in axonal mitochondria size and motility.**
**(A)** Representative images of mitochondrial morphology and distribution in larval axons. Mitochondria were detected using mitoGFP in controls (*LacZ*-RNAi) and *pdzd8*-RNAi expressing larvae. Scale bar = 5 μm. **(A, B, C)** Mitochondrial length (B), and mitochondrial density (C) in the larval axons shown in (A) were analysed using ordinary one-way ANOVA and Holm–Sidak's multiple comparisons. n = 10, 10, 13 animals, data points represent different axons, all differences ns. **(D)** Representative kymographs showing motility of mitoGFP signal in controls and *pdzd8*-RNAi–expressing larvae. Stationary mitochondria appear as vertical lines, moving mitochondria form diagonal lines in anterograde or retrograde directions. **(D, E)** Quantification of mitochondrial transport shown in (D), analysed using ordinary one-way ANOVA and Holm–Sidak's multiple comparisons, n = 14–25 larvae, P < 0.0001. **(F)** Representative images of NMJs and mitochondria of controls and *pdzd8*-RNAi labeled using mitoGFP. magenta = mitoGFP, green = anti-HRP (neuronal membrane). Scale bar = 10 μm. **(G, H, I)** Quantifications of total volume, P = 0.0036, (G), mitochondrial volume, P = 0.002 (H) and mitochondrial density (I) of 16–20 NMJs were compared using an unpaired t test with Welch's Correction.

dramatically delayed age-associated decline in locomotor activity and significantly extended median lifespan compared with control animals. Our results corroborate a recent study showing that adult *Pdzd8*-deficient mice display increased locomotor activity compared with control littermates (Al-Amri et al, 2022).

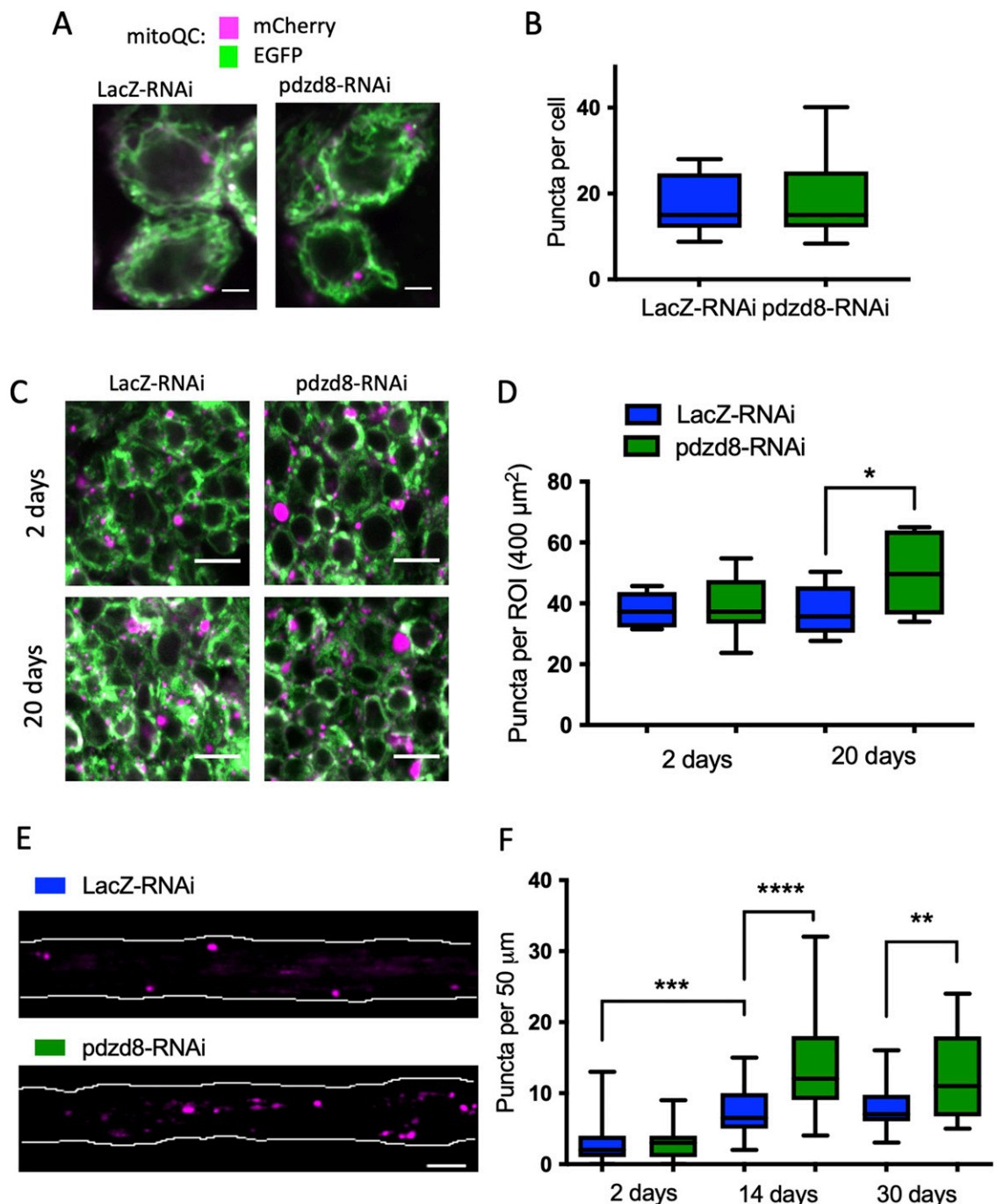

**Figure 5.  Pan-neuronal *pdzd8*-RNAi increases mitophagy during aging.**
**(A)** Representative images of MitoQC signal in wandering L3 larval ventral ganglia. magenta = mCherry, green = GFP, images show a single plane of a Z stack. Scale bar = 2 μm. **(A, B)** Quantification of MitoQC puncta shown in (A), n = 9 ROIs, differences ns. **(C)** Representative images of MitoQC signal in adult brains in 2- and 20-d-old flies, magenta = mCherry, green = GFP. Scale bar = 5 μm, image shows a single plane of a Z stack. **(D, E)** Quantification of MitoQC signal in adult brains and compared using an unpaired *t* test with Welch's correction (n = 7–9 ROIs, *P* = 0.0072) (E) the Representative images of MitoQC signal in 14-d-old fly wings. Only mCherry signal (magenta) is shown for clarity. Wing nerve are outlined (white). Scale bar = 5 μm. **(F)** Quantification of MitoQC signal in aged fly wings at 2, 14, and 30 d post-eclosion using a one-way ANOVA with Holm–Sidak's multiple comparisons. n (2 d) = 33, 26 wings (14 d) = 24, 31 wings (30 d) = 32, 12 wings. ***P* < 0.01, ****P* < 0.001, *****P* < 0.0001.

Because decline in mitochondrial transport is proposed to contribute to neuronal aging (Vagnoni & Bullock, 2018), we hypothesized that reducing tethering in the aging flies might be protective by allowing sustained mitochondrial motility (Mattedi &

Vagnoni, 2019). However, we detected no change in the percentage of motile mitochondria in larval neurons with reduced *pdzd8* expression. Because knockdown of *pdzd8* also prolonged the survival of flies fed mitochondrial toxins, this suggests that the protective

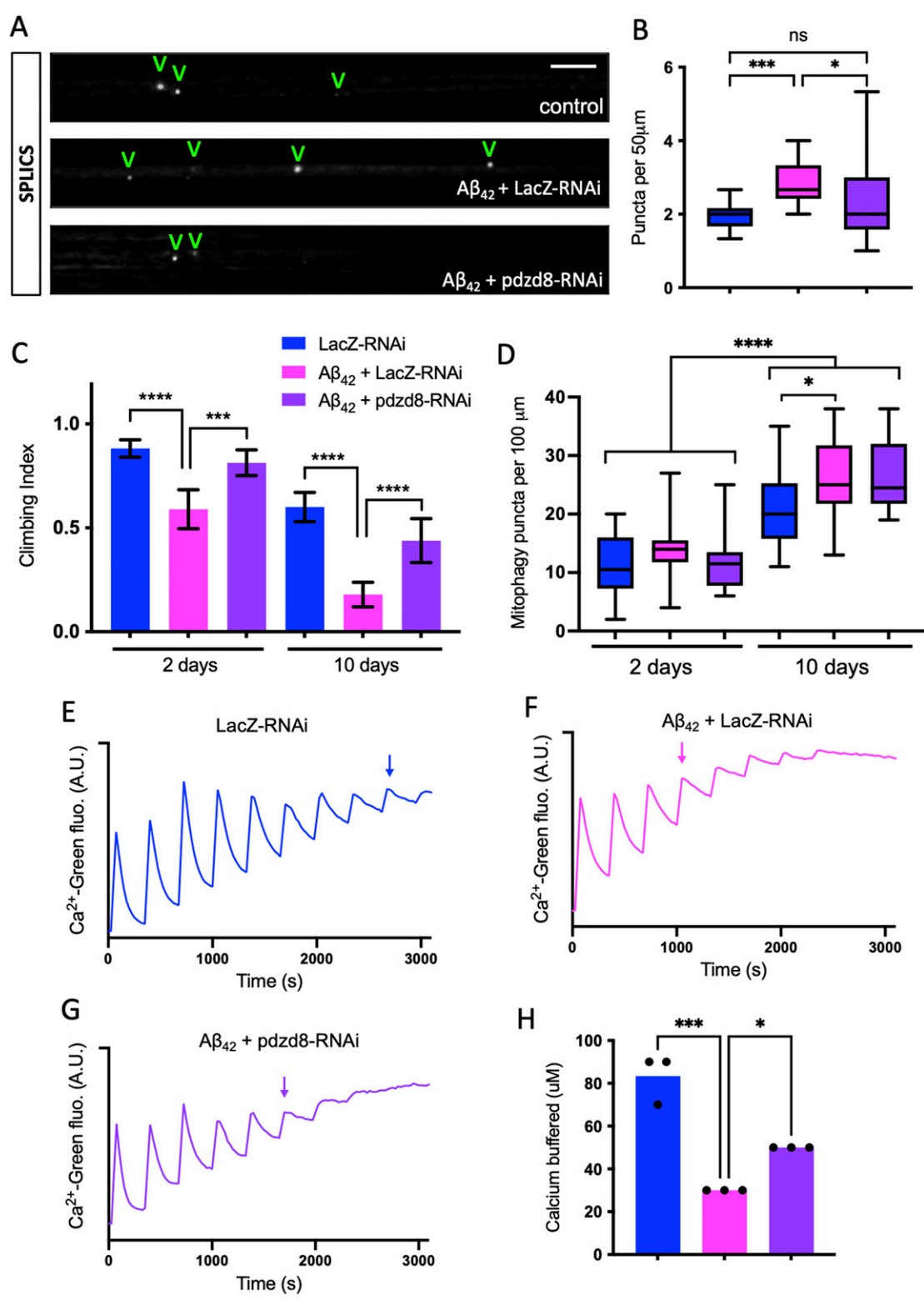

**Figure 6.  Reducing pdzd8-mediated MERCs rescues the locomotor defects in an Alzheimer's disease model.**
**(A)** SPLICS puncta indicating MERCs in axon bundles of larval neurons from $A\beta_{42}$ expressing flies compared with controls. Quantified puncta highlighted with V.
**(A, B)** Quantification of SPLICS signal in (A) using Kruskal–Wallis test with Dunn's post hoc correction, n = 6 larvae per genotype, three ROIs averaged per larva.
**(C)** Locomotor (climbing) activity of flies of the indicated ages, comparing control versus $A\beta_{42}$ with control or *pdzd8*-RNAi. n > 65 flies. Statistical analysis was performed using Kruskal–Wallis test with Dunn's post hoc correction. **(D)** Quantification of MitoQC signal in aged fly wings at 2 and 10 d post-eclosion using Kruskal–Wallis test with Dunn's post hoc correction, n = 14 wings. **(E, F, G)** Representative recordings of mitochondrial calcium retention capacity, monitoring extramitochondrial Calcium-Green

effects of reducing MERCs might instead result from more efficient clearance of damaged mitochondria.

Clearance of damaged mitochondria via mitophagy is also thought to be a key factor in healthy aging of neurons (Pickrell & Youle, 2015; Whitworth & Pallanck, 2017; Ma et al, 2018). It remains unclear, however, whether increased or decreased mitophagy in neurons is protective during aging (Montava-Garriga & Ganley, 2020) and if specific neuronal populations are more susceptible to mitochondrial turnover. Here we provide the first in vivo evidence that mitophagy may be regulated by pdzd8-mediated MERCs. Mitophagy levels did not change in young flies with less pdzd8, but when aged, these flies displayed significant increases in mitophagy in both soma and axons.

MERCs mediated by pdzd8 may limit the rate of mitophagy, analogous to the protective role that mitochondrial fusion is thought to play during starvation-induced autophagy (Gomes et al, 2011; Rambold et al, 2011; Rana et al, 2017). Consistent with this, increasing several MERC proteins can slow toxin-induced mitophagy in non-neuronal cultured cells (McLelland et al, 2018; McLelland & Fon, 2018), and MERCs might directly regulate mitophagy in mammalian neurons (Puri et al, 2019). Pdzd8 has also been found to mediate contacts between the ER and lysosomes (Guillen-Samander et al, 2019) and three way contacts between the ER, mitochondria and late endosomes (Elbaz-Alon et al, 2020; Shirane et al, 2020). Pdzd8 function at any of these contact sites could potentially alter mitophagy and future work should explore the mechanisms behind our observed increases in mitophagy.

Altered MERCs have been reported in patients and cell models of AD (Area-Gomez et al, 2012). We also observed excess MERCs in a fly model of AD which was reversed by knockdown of pdzd8. Importantly, reduction of pdzd8 was sufficient to significantly reduce their age-associated decline in climbing. Although boosting mitophagy has been shown to be protective in worm and mouse AD models (Fang et al, 2019), our data indicate that this is not further up-regulated in the AD flies upon loss of pdzd8 so unlikely to account for the rescue. In contrast, we found that the CRC was much reduced in the AD flies, consistent with excess calcium transfer from excess MERCs, and this was partially restored by reduction in pdzd8.

In summary, we propose that reducing pdzd8-mediated MERCs may be protective in aging neurons via a number of mechanisms including increased turnover of damaged mitochondria and reducing excess mitochondrial calcium buffering. As regulators of multiple aspects of mitochondrial biology, manipulating MERCs may provide an avenue for enhanced mitochondrial homeostasis to help promote healthy aging of neurons.

# Materials and Methods

## Husbandry

Flies were raised under standard conditions at 25°C on food containing agar, cornmeal, molasses, malt extract, soya powder, propionic acid, nipagin, and yeast in a 12 h:12 h light:dark cycle.

## Genetics

*Drosophila* lines used in this study were obtained as indicated in Table S1, or generated as described below. Detailed genotypes for each sample are shown in Table S2. All mutant lines used in this study were backcrossed to an isogenic $w^{1118}$ strain (RRID: BDSC_6326), for four to six generations before use. For all integration events, multiple independent lines were initially isolated, verified by PCR and assessed for consistent effects before selecting a single line of each integration site for further study. Wherever possible, inert UAS lines such as UAS-*LacZ*-RNAi and UAS-*mito-Cherry* are used as dilution controls to ensure equal numbers of UAS constructs in control and experimental conditions. Unless otherwise stated, male flies were used in all experiments.

## New lines

### SPLICS

The SPLICSs construct with an 8–10 nm range from Tito Cali & Marisa Brini (Cieri et al, 2017), was amplified from pSYC-SPLICSs-P2A using (TAAGCAGCGGCCGCTGATTTAGGTGACACTATAG) and T7 forward primer (TAATACGACTCACTATAGGG) and cloned into pUAST.attB between NotI and XbaI sites. Flies were injected by BestGene to insert into attP16 (II) and attP2 (8622, III) and the attP16 site gave better signal and so was used in this work. The number of puncta produced in the axon bundles driven by nSyb-GAL4 varied with a consistently more puncta than in the central axons bundles in the peripheral bundles (Fig S2B).

### pdzd8-HA

*pdzd8*-HA was synthesized by Genewiz based on the cDNA GenBank sequence LD34222 (AY118553.1) (Sayers et al, 2020), and cloned into pUAST.attB between EcoRI and XbaI. The University of Cambridge Department of Genetics Fly Facility generated lines by injection of this construct into the attP40 landing site.

## Climbing

The startle induced negative geotaxis (climbing/locomotor) assay was performed as described previously (Andreazza et al, 2019). Briefly, a maximum of 23 males were placed into the first tube of a counter-current apparatus, tapped to the bottom, and given 10 s to climb 10 cm. This procedure was repeated five times (five tubes), and the number of flies that has remained into each tube counted and the climbing performance expressed as a climbing index (Greene et al, 2003). The same flies were aged and assayed again on the indicated days post-eclosion.

## Lifespan

For lifespan experiments, groups of ~20–25 males were collected with minimal time under anesthesia ($CO_2$), placed into separate vials with food and maintained at 25°C. Flies were transferred into

---

5N fluorescence levels (AU), of mitochondria from flies of the indicated genotypes. Arrows indicate calcium pulse where retention capacity is exceeded. **(E, H)** Quantification of calcium retention capacity in (E). n = 3, one-way ANOVA with Dunnett's correction. *$P < 0.05$, **$P < 0.01$, ***$P < 0.001$, ****$P < 0.0001$.

fresh vials every 2–3 d, and the number of dead flies were recorded. Percent survival was calculated using https://flies.shinyapps.io/Rflies/. To assess lifespan in a diet with restricted nutrients, flies were raised in standard conditions then transferred to tubes containing food made from 5% sucrose and 1% agar and flipped every 2–3 d. Lifespans in the presence of mitochondrial toxins and hydrogen peroxide were also performed on food made from 5% sucrose and 1% agar cooled to less than 50°C before adding toxin at 1:1,000. Rotenone (R8875; Sigma-Aldrich) was dissolved in DMSO (1 mM final concentration) and antimycin A (A8674; Sigma-Aldrich) (4 $\mu$g/ml final concentration) dissolved in 70% ethanol. Flies in toxin assays were starved for 5 h before being placed on food containing toxins. Flies in rotenone assays were monitored twice a day and flipped every 2 d. Flies in antimycin A assays were monitored three times a day and flipped every 2 d.

## Fluorescence microscopy

Imaging of larval axons was performed as described by Wang and Schwarz (2009) with the following variations: wandering third instar larvae were pinned at each end dorsal side up to a reusable Sylgard (761028; Sigma-Aldrich) coated slide using pins (FST26002-10; Fine Science Tools) cut to ~5 mm and bent at 90°. The larvae were cut along the dorsal midline using micro-dissection scissors. Internal organs were removed with forceps without disturbing the ventral ganglion and motor neurons. Larvae were then covered in dissection solution (Godena et al, 2014). The cuticle was then pulled back with four additional pins. The anterior pin was adjusted to ensure axons are taut and as flat as possible for optimal image quality.

Movies were taken using a Nikon E800 microscope with a 60× water immersion lens (NA 1.0 Nikon Fluor WD 2.0) and an LED light source driven by Micromanager 1.4.22 Freeware (Edelstein et al, 2014). A CMOS camera (01-OPTIMOS-F-M-16-C) was used to record 100 frames at a rate of 1 frame per 5 s (8 min 20 s total). Axons were imaged within 200 $\mu$m of the ventral ganglion in the proximal portion of the axons and no longer than 1 h after dissection. Movies were converted into kymographs using Fiji (Schindelin et al, 2012) and mitochondrial motility quantified manually with the experimenter blinded to the condition.

For SPLICS imaging in axon bundles, at least three ROI 50 × 12 $\mu$m were quantified per animal and averages for each larva were plotted. For SPLICS quantification puncta intensity varied considerably, so blinded manual counting was used.

To image NMJs, larvae were dissected as described above and fixed for 20 min in 4% formaldehyde in PBS. After blocking for 1 h in 1% BSA/0.3% Triton X-100/PBS solution, anti-HRP was added at 1:500 and samples agitated gently overnight at 4°C. After three washes in 0.3% Triton X-100/PBS at room temperature, samples were incubated with Alexa Fluor 594 at 1:500 for 1 h in 1% BSA/0.3% Triton X-100/PBS solution. Samples were then washed in 3× in PBS before being mounted in Prolong Diamond. NMJs were imaged on a Nikon Eclipse TiE inverted microscope with appropriate lasers using an Andor Dragonfly 500 confocal spinning disk system, using an iXon Ultra 888 EMCCD camera (Andor), coupled with Fusion software (Andor) using a 60× NA 1.49 objective. NMJs on muscle 4 from segments A3 and A4 (NMJs on these segments are the same size

[Nijhof et al, 2016]) were captured in Z stacks with 0.3 $\mu$m step size and analysed using Imaris (x64 9.2.0) to determine NMJ volume, mitochondrial volume, and mitochondrial number.

For MitoQC imaging, samples were fixed for 30 min in 4% formaldehyde (16% 100503; VWR) diluted in pH 7.0 PBS. Adult brains were mounted in Prolong Diamond Antifade Mountant (P36961; Thermo Fisher Scientific) using spacers and imaged on a Carl Zeiss LSM880 confocal laser-scanning system on an Axio Observer Z1 microscope (Carl Zeiss), coupled with ZEN software (Carl Zeiss) using a 100× Plan-APOCHROMAT/1.4 oil DIC objective. Images are shown in false colour with magenta puncta representing mCherry signal indicating where the reporter is in an acidic environment of a lysosome and the GFP has been quenched (Allen et al, 2013).

Imaging of wings was performed as described by Vagnoni and Bullock (2016). Briefly, flies were anaesthetized with $CO_2$ and immobilised with their wings outstretched on a cover glass with a fine layer of Halocarbon oil (VWR). A second cover glass was then added on top of the fly to stabilize the sample. Live imaging in the wing nerves was performed using a Nikon spinning disk system essentially as described previously (Morotz et al, 2019). The mitoQC puncta were annotated with Fiji using the Cell Counter plugin and quantified with the experimenter blinded to the genotype.

For imaging of the larval epidermal cells, the larvae were dissected as described above, but the nervous system was also removed before fixation. The samples were washed in PBS and the muscles were then removed (Tenenbaum & Gavis, 2016). The dissected filets were mounted in Prolong Diamond Antifade Mountant using No. 1.5H High Precision Deckglaser cover slips and placed under a weight for 24 h. Nikon Structured Illumination Microscopy imaging was performed on a Nikon Ti Eclipse with an Andor DU-897 X-5835 camera and SR Apo TIRF 100× (NA1.5) objective run using NIS-Elements 4.60. Images were analysed in Fiji (Schindelin et al, 2012) using the Coloc2 plugin.

## Transmission electron microscopy

TEM was performed at Cambridge Advanced Imaging Center (CAIC). Brains of 2-d-old adult flies were fixed in 2% glutaraldehyde/2% formaldehyde in 0.1 M sodium cacodylate buffer, pH 7.4, containing 2 mM $CaCl_2$ and 0.1% Tween 20 (based on method described in Celardo et al [2016]), overnight at 4°C. Samples were then washed 5× with 0.1 M sodium cacodylate buffer and then treated with osmium for 2 d at 4°C (1% $OsO_4$, 1.5% potassium ferricyanide, and 0.1 M sodium cacodylate buffer, pH 7.4). Samples were then washed 5× in distilled water and treated with 0.1% aqueous thiocarbohydrazide for 20 min in the dark at room temperature. Samples were washed another 5× in distilled water then treated with osmium a second time for 1 h at room temperature (2% $OsO_4$ in distilled water). Samples were then washed another 5× in distilled water before being treated with uranyl acetate bulk stain for 3 d at 4°C (2% uranyl acetate in 0.05 M maleate buffer pH 5.5). After a final 5× wash in distilled water, samples were dehydrated in 50/70/95/100% ethanol, 3× in each for at least 5 min each. Dehydration was completed by two further treatments with 100% dry ethanol, 2× in 100% dry acetone, and 3× in dry acetonitrile for at least 5 min each. Quetol resin mix (12 g Quetol 651, 15.7 g NSA, 5.7 g MNA, 0.5 g benzyldi-methylamine) made with an equal volume of 100% dry acetonitrile

and samples placed in this mix for 2 h at room temperature. Samples were then incubated in pure Quetol resin mix for 5 d, exchanging the samples to fresh resin mix each day. After 5 d, the brains were embedded in coffin moulds and cured at 60°C for at least 48 h. Ultrathin sections were cut on a Leica Ultracut E at 70-nm thickness. Sections were mounted on 400 mesh bare copper grids and viewed in a FEI Tecnai G20 electron microscope run at 200 keV using a 20-$\mu$m objective aperture. Images were taken in the cell bodies of the posterior protocerebrum where the organelle morphology was most distinct. Quantification of the percentage of clearly identifiable mitochondria in contact with ER was performed manually as described by Celardo et al (2016), and the experimenter was blinded to the genotype.

### qPCR

Five female wandering third instar larvae per sample were washed briefly in 1× PBS, placed in RNAse free tubes and frozen on dry ice. Larvae were homogenized in Trizol and RNA isolated by phenol:chloroform extraction and isopropanol precipitation. DNAse treatment using Invitrogen TURBO DNA-free rigorous procedure was performed before measuring RNA concentration with a Qubit RNA HS Assay Kit (Molecular Probes, Life Technologies). Reverse transcription reactions used 1.32 $\mu$g of RNA using SuperScript III Reverse Transcriptase (Invitrogen) with Oligo(dT)23VN as per manufacturer's instructions. The resulting cDNA was used for qPCRs using PowerUp SybrGreen (A25742; Applied Biosystems). Primers for pdzd8 were PDZD8-F TTCTGTTTGGCTTCTCCTGG, PDZD8-R TTGAGGAACTGCGACTGATC designed using RealTime qPCR Assay Entry (https://www.idtdna.com). *αTub84B* (Fwd: TGGGCCCGTCTGGACCACAA, Rev: TCGCCGTCACCGGAGTCCAT), *vkg* (Fwd: CGAGGATGTTACCCAGAGATC, Rev: TGCGTCCCTTGATTCCTTTG), *COX8* (Fwd: CAGAGCCGTTGCCAGTC, Rev: CTTGTCGCCCTTGTAGTCC), and *Rpl32* (Fwd: AAACGCGGTTCTGCATGAG, Rev: GCCGCTTCAAGGGACAGTATCTG) were used as housekeeping genes with their values combined to compare knockdown with the geometric mean (Taylor et al, 2019).

### ATP

ATP levels were measured in 2- and 20-d-old fly heads with 40 flies per genotype and three biological replicates. The ATP levels were measured as described by Tufi et al (2019) with minor modifications. Briefly, heads were homogenized in 6 M guanidine-Tris/EDTA extraction buffer and subjected to rapid freezing in liquid nitrogen. Luminescence produced from homogenates mixed with the CellTiter-Glo Luminescent Cell Viability Assay (Promega) was measured with a SpectraMax Gemini XPS luminometer (Molecular Devices) and normalized to total protein, quantified using the Pierce BCA method (Thermo Fisher Scientific).

### Calcium retention capacity

Mitochondrial isolation and calcium flux methods were adapted from Tufi et al (2019). Briefly, 50 whole adult flies (day 5) were homogenized using a 2 ml Wheaton Dounce Tissue Grinder and with a loose-fitting pestle in 400 $\mu$l of mitochondrial isolation buffer (225 mM mannitol, 75 mM sucrose, 5 mM Hepes, 0.1 mM EGTA, pH 7.4, and

2% BSA). The homogenates were spun at 1,500$g$ at 4°C for 6 min before being filtered through a 70 $\mu$m nylon cell strainer (352350; Falcon). The filtrate was centrifuged at 7,000$g$ at 4°C for 6 min and the resulting pellet was resuspended in 200 $\mu$l mitochondrial isolation buffer without BSA. Protein concentration was quantified using a BCA assay and the mitochondria were resuspended at 1 mg/ml in assay buffer (250 mM sucrose, 10 mM MOPS-Tris, 5 mM/2.5 mM glutamate/malate-Tris, 5 mM Pi-Tris, 10 $\mu$M EGTA, and 1 $\mu$M Calcium Green 5N, pH 7.4). The fluorescent intensity of Calcium Green was measured kinetically at 25°C using the CLARIOstar Plus (BMG LABTECH) (Ex. 485, Em. 530). The mitochondria were pulsed with 10 $\mu$M CaCl$_2$ every 5 min for a total of 50 min. The CRC was determined by the total concentration of CaCl$_2$ successfully buffered by the isolated mitochondria where buffering is considered successful if the extramitochondrial calcium level has been returned to 50% of the level before addition.

### Quantification and statistical analysis

Statistical analyses were performed using GraphPad Prism 9 software. Data are reported as mean ±95% CI unless otherwise stated in figure legends. Climbing was assessed using a Kruskal–Wallis non-parametric test with Dunn's post hoc correction for multiple comparisons. Lifespans were compared using log-rank Mantel–Cox tests. Number of flies and *P*-values are reported in the figure legends.

Mitochondrial transport was analysed using ordinary one-way ANOVA and Holm–Sidak's multiple comparison. ATP measurements were analysed by two-tailed *t* test. Values are not significantly different to controls unless otherwise stated.

SCope (http://scope.aertslab.org/) was used to visualize transcriptome data from the unfiltered adult fly brain dataset (Davie et al, 2018).

## Supplementary Information

## Acknowledgements

We thank Vinay Godena, Caspar Baldwin, Alvaro Sanchez-Martinez, Tom Gleeson, Juliette Lee, and Wing Hei Au for help with various aspects of the fly work; Ana Terriente-Felix for help with stock maintenance and generating lines; Luis Gracia for developing the lifespan data analysis tool (https://flies.shinyapps.io/Rflies/); David Pate and Steve Drinkwater for all their help setting up our fly lab; Carlo Viscomi and Hiran Prag for useful critical discussion; Richard Mann and Sumaira Zamurrad for help with completing the fly work; Cristiane Benincà, Jordan Morris, and the Wohl Cellular Imaging Centre at King's College London for their help with microscopy; Karin H Muller, Lyn Carter, and Filomena Gallo for their help with TEM at the Cambridge Advanced Imaging Center (CAIC); and Jane Stinchcombe and Sam Loh for their help in interpreting this data. We thank Megan Oliva and Erin Barnhart and her laboratory for their critical reading of the manuscript. We thank Isabel Palacios for the A$\beta$-Arctic flies and Luca Scorrano and Valentina Debattisti for the synthetic tether line. Other stocks were obtained from the Bloomington Drosophila Stock Center which is supported by grant National

Institutes of Health (NIH) P40OD018537. This work was supported by Medical Research Council core funding (MC_UU_00015/6 and MC_UU_00028/6 to AJ Whitworth; MC_UU_00015/7 and MC_UU_00028/5 to J Prudent) and European Research Council Starting grant (DYNAMITO; 309742) to AJ Whitworth; a NC3Rs David Sainsbury fellowship and Skills and Knowledge Transfer grant (N/N001753/2 and NC/T001224/1), an Academy of Medical Sciences Springboard Award (SBF004/1088), and a van Geest Fellowship in Dementia and Neurodegeneration and van Geest PhD studentship awards to A Vagnoni. VL Hewitt was funded by an EMBO Long-Term Fellowship (ALTF 740-2015) and co-funded by the European Commission FP7 (Marie Curie Actions, LTFCOFUND2013, GA-2013-609409). Stocks were obtained from the Bloomington Drosophila Stock Center which is supported by grant NIH P40OD018537.

## Author Contributions

VL Hewitt: conceptualization, data curation, formal analysis, investigation, methodology, and writing—original draft, review, and editing.
L Miller-Fleming: data curation, investigation, methodology, and writing—review and editing.
MJ Twyning: data curation, formal analysis, investigation, methodology, and writing—review and editing.
S Andreazza: data curation, formal analysis, investigation, methodology, and writing—review and editing.
F Mattedi: data curation, formal analysis, investigation, methodology, and writing—review and editing.
J Prudent: supervision, funding acquisition, investigation, and writing—review and editing.
F Polleux: supervision, funding acquisition, investigation, and writing—review and editing.
A Vagnoni: data curation, formal analysis, funding acquisition, investigation, methodology, and writing—review and editing.
AJ Whitworth: conceptualization, data curation, formal analysis, supervision, funding acquisition, and writing—original draft, review, and editing.

## Conflict of Interest Statement

The authors declare that they have no conflict of interest.

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
