## [Reviewer comments · Life Science Alliance]

Life Science Alliance

Decreasing pdzd8-mediated mito-ER contacts improves organismal fitness and mitigates A β 42 toxicity

Victoria Hewitt, Leonor Miller-Fleming, Madeleine Twyning, Simonetta Andreazza, Francesca Mattedi, Julien Prudent, Franck Polleux, Alessio Vagnoni, and Alexander Whitworth

DOI: 10.26508/lsa.202201531

Corresponding author(s): Alexander Whitworth, University of Cambridge

Review Timeline:

Submission Date:	2022-05-24
Editorial Decision:	2022-06-30
Revision Received:	2022-07-01
Accepted:	2022-07-01

Transaction Report:

Please note that the manuscript was reviewed at Review Commons and these reports were taken into account in the decision-making process at Life Science Alliance.

June 30, 2022

RE: Life Science Alliance Manuscript #LSA-2022-01531

Dr. Alexander J Whitworth
Medical Research Council
Mitochondrial Biology Unit
Hills Road
Cambridge CB2 0XY
United Kingdom

Dear Dr. Whitworth,

Thank you for submitting your revised manuscript entitled "Decreasing pdzd8-mediated mitochondria-ER contact sites in neurons improves organismal fitness and mitigates A β 42 toxicity". We would be happy to publish your paper in Life Science Alliance pending final revisions necessary to meet our formatting guidelines.

- please upload the main manuscript text as an editable doc file
- please upload your figures as single files (both the main and supplementary figures)
- please add a running title, summary blurb, category, and add the Twitter handle of your host institute/organization as well as your own or/and one of the authors in our system
- please add a callout for your tables in your main manuscript text

A. FINAL FILES:

B. MANUSCRIPT ORGANIZATION AND FORMATTING:

**Submission of a paper that does not conform to Life Science Alliance guidelines will delay the acceptance of your

manuscript.**

The license to publish form must be signed before your manuscript can be sent to production. A link to the electronic license to publish form will be sent to the corresponding author only. Please take a moment to check your funder requirements.

Sincerely,

Reviewer #1 (Comments to the Authors (Required)):

Dr. Alexander J Whitworth and his colleagues should be commended on their well-conducted new experiments and engaging new manuscript. The new manuscript contains a big amount of data that are interpreted to strongly support their conclusion that interrupting neuronal pdzd8-mediated mitochondria-ER contact sites improves organismal fitness and mitigates A β 42 toxicity. This reviewer has no additional comments.

Response to Reviewers

We thank the reviewers for their careful reading of our manuscript and their valuable suggestions and comments. To address the reviewers' concerns and improve our manuscript, we have completed the additional experiments and revised the text as described below.

Reviewer #1

(Evidence, reproducibility and clarity (Required)):

****Summary:****

Provide a short summary of the findings and key conclusions (including methodology and model system(s) where appropriate). Please place your comments about significance in section 2.

The authors present an in vivo analysis of pdzd8 (CG10362) and a synthetic ER-mitochondria tether in the regulation of locomotor activity, lifespan, and mitochondrial turnover of *Drosophila melanogaster*, using basic bioinformatics, RNAi, SPLICS, imaging and microscopies observations (i. e. TEM, SIM), fly lines, and a representative AD fly disease model, etc. The research methodologies were detailed in good order. The model system employed was suitable to address the research topic. The manuscript was written in a clear language and statistical analysis were correctly applied.

****Major comments:****

-Are the key conclusions convincing?

Yes. The results/conclusions are logical and provide an overview of Pdzd8 in the regulation of mitochondrial quality control and neuronal homeostasis.

-Would additional experiments be essential to support the claims of the paper? Request additional experiments only where necessary for the paper as it is, and do not ask authors to open new lines of experimentation.

No. Experiments were generally well performed, and all the data support the conclusions.

-Are the suggested experiments realistic in terms of time and resources? It would help if you could add an estimated cost and time investment for substantial experiments.

No suggested experiments needed.

-Are the data and the methods presented in such a way that they can be reproduced?

Yes. The authors have followed proper experimental design and methods have been described in sufficient detail.

-Are the experiments adequately replicated and statistical analysis adequate?

Yes, they are.

Minor comments:

-Specific experimental issues that are easily addressable.

No comment.

-Are prior studies referenced appropriately?

Yes. The relevant literatures have been cited appropriately.

-Are the text and figures clear and accurate?

1. Please pay attention to the correct spelling of the described protein name (Pdzd8) and gene name (should be in 'italic') throughout the manuscript, i. e. line 36, 98, and 556, etc.

As this is the first characterization of the fly homolog of the mammalian Pdzd8 We have decided to name the fly protein "pdzd8", using the lower case "p" to distinguish it from the mammalian protein, in line with common practice. We have checked and corrected our use of italics for the gene name as noted in track changes.

2. In figure 1C and its figure legend, please state what the numbers "201" and "195" stand for. We have added the text "numbers on bars indicate number of mitochondria analysed" to the figure legend.

3. Your data needs to be converted the lowercase letter "x" to math symbol " \times " when representing times sign, i. e. line 523, 5x, etc.

Corrected

-Do you have suggestions that would help the authors improve the presentation of their data and conclusions?

No comment.

Reviewer #1

(Significance (Required)):

-Describe the nature and significance of the advance (e.g. conceptual, technical, clinical) for the field.

Discoveries from this study include 1) characterization of the tethering protein Pdzd8 in *Drosophila melanogaster*, and 2) shed light on a possible way on how to enhance mitochondrial quality control and to help promote healthy aging of neurons by manipulating MERCs.

-Place the work in the context of the existing literature (provide references, where appropriate).

With this manuscript, the authors present a straightforward but sound piece of scientific

research, with the intent to illustrate the consequences of neuronal depletion of pdzd8 in *Drosophila melanogaster*. Since Pdzd8 plays specific functions in ER-mitochondrial tethering complexes and dysregulations of MERCs are damaging to neurons, this protein represents a good potential target. In this context the characterization of Pdzd8 should represent an interesting starting point. To this purpose, the gene was knockdown and the tether construct was recombinantly produced. The fly lines were then subjected to analysis both at the organismal and at the cellular level.

-State what audience might be interested in and influenced by the reported findings.

Audience might include those who are in the field of neuroscience and pharmaceutical, and benefit from an awareness of this research.

-Define your field of expertise with a few keywords to help the authors contextualize your point of view. Indicate if there are any parts of the paper that you do not have sufficient expertise to evaluate.

Key words in my field of expertise: Ageing, neurodegenerative diseases, Alzheimer's disease, mitophagy, NAD+, neuroprotection.

My group is investigating the molecular mechanisms of ageing and age-related neurodegeneration (especially AD) using cross-species model systems, ranging from human brain samples, iPSCs, *C. elegans*, *Drosophila melanogaster*, and mice, therefore I have sufficient expertise to evaluate this paper.

Referees Cross-commenting

To this reviewer the key novelty of this paper was the study of the regulation of the mitochondrial-ER contact sites (MERCs) in life and health. The data indicate that MERCs mediated by the tethering protein pdzd8 play a critical role in the regulation of mitochondrial homeostasis, neuronal function, and lifespan. In a transitional perspective, this reviewer would ask to check whether this mechanism conserves in rodents or not (e.g. to memory in the AD mice and to run lifespan in mitochondrial toxin condition). This may be too much. But will depend on the standard of the journal.

We thank the reviewer for their input, evaluation and interest. We too are keen to know whether this mechanism is conserved and hope to investigate this in our ongoing work including characterizing a mouse mutant, but the current work already represents a substantial investment of resources and a worthy study in its own right as the first description of the *in vivo* role of pdzd8, so we feel it is beyond the scope of the current work.

Reviewer #2

(Evidence, reproducibility and clarity (Required)):

Hewitt et al. describe and characterize for the first time the ortholog of pdzd8 in *Drosophila melanogaster*. In accordance with pdzd8's previously described function as a member of mitochondrial-ER contact sites (MERCs) the authors show reduced MERCs upon RNAi mediated depletion of pdzd8 via TEM, SIM and a split-GFP based contact site sensor. Pdzd8 depletion results in the increased life span as well as improved locomotor activity in aging flies while increase of MERCs with a synthetic tether accelerates the age-related declines in survival and locomotion. Moreover, pdzd8 depleted flies are more resistant against mitochondrial toxins. The authors correlate these protective effects of pdzd8 knockdown with an increase in

mitophagy using a mitophagy sensor and describe a rescue of locomotor defects in an Alzheimer disease fly model by *pdzd8* depletion.

Major comments:

1. The authors quantify the number of MERCs in thin sections of TEM (Fig 1B and C). It would add to the paper if the authors would show a representative reconstruction of the quantified somata, as a 3D reconstruction would visualize ER-Mito contacts more reliable than thin sections.

We agree that the 3D reconstruction of TEM images would extend the current analyses, however such advanced techniques are not readily available, and the samples used to collect the TEM data are not suitable for 3D reconstructions. To counter this, we have used three independent methods to analyse the changes in MERCs, all of which show a decrease in MERCs in the flies with *pdzd8* knockdown, supporting that these observations are reproducible and robust.

2. The authors quantify MERCs in *pdzd8* KD also by SIM (Fig 1F, G). However, they quantify the number of MERCs in epidermal cells while they also show SIM images of larval neurons (Fig S1D). For consistency and to support their claim of MERC reduction in neurons, we ask the authors to include the quantification based on larval neurons especially as the authors show that *pdzd8* is predominantly expressed in the CNS.

Unfortunately, the soma of larval neurons have extremely limited cytosol (see Fig. S1D) which creates very challenging conditions to discern the spatial separation of ER and mitochondria by light microscopy. While co-localisation of organelle markers in such cells has been reported in the literature, we are dubious that the restricted space within the cytosol will allow reliable analysis of spatial resolution in these cells. In contrast, epidermal cells are much larger providing greater spatial separation of ER and mitochondria. Notably, we complemented the co-localisation analysis of epidermal cells with two additional approaches, TEM analysis and the SPLICS reporter construct, to demonstrate *pdzd8*-RNAi results in decreased MERCs specifically in neurons.

3. The authors describe a decreased NMJ volume in Fig 4G. It would improve and complete the functional characterization of *pdzd8* in flies if the authors can provide further data whether *pdzd8* KD causes a general synaptic defect. Can the authors show morphological synaptic defects in the existing TEM data of the adult brain or provide additional ERG recordings, which would elucidate the functional consequences of *pdzd8* depletion in the CNS?

Our TEM data are not suitable for us to properly analyse defects in synaptic morphology as our images centered around the cell bodies where the organelle morphology was easiest to distinguish and there are very few synapses. While it is not surprising that the knockdown of *pdzd8* has some detrimental effects, we chose to focus our efforts on trying to determine the cause of the protective effect at the organism level, i.e., on locomotor activity in aged flies, rather than to exhaustively characterise the myriad phenomena which may be impacted as a knock-on effect of the disrupted cell biology that we have demonstrated. We hope to further explore the detrimental functional consequences of *pdzd8* depletion on such phenomena as neurotransmission in future work but feel that this analysis is beyond the scope of this study.

4. Hewitt et al. suggest a beneficial effect of increased turnover of mitochondria for healthy aging. To convince readers we would like to ask the following:

a) This claim is based on their observation of increased mitophagy in *pdzd8* depleted flies using

one reporter (Fig 5). Can the authors support their data with an alternative method as this is one of the key claims of the manuscript?

We appreciate the reviewer's comment here. We consider that the mitophagy reporters are now reasonably well-established and relied on in the field. In our hands, we have found the mito-QC to perform better and more consistent than mt-Keima as previously documented (Lee et al. 2018 JCB doi: 10.1083/jcb.201801044). Nevertheless, our subsequent analysis on mitophagy, prompted by the query below, has meant that we have re-evaluated our interpretation and consequently downplayed the contribution of increased mitophagy to the rescue mechanism.

b) An increased turnover of Mitochondria would also suggest that there are more "young" mitochondria present in the *pdzd8* KD neurons. Can the authors experimentally address that? We understand the reviewer's point here but due to the continual fission and fusion, as well as piecemeal turnover of mitochondria (see Vincow et al. 2019 Autophagy doi: 10.1080/15548627.2019.1586258), we consider that the concept of 'young' versus 'old' mitochondria is misplaced. The mitochondrial network essentially exists as a milieu of components which are produced and degraded at different rates.

c) Furthermore, we would like to ask the authors to use also the MERC tether as control in the mitophagy assay. This would allow further conclusions about the role of the mitophagy, its protective effect during aging and the role of MERCs in this process.

Unfortunately, this MERC tether is constructed from an RFP with N- and C-terminal tethering peptides. The presence of this RFP abrogates the proper analysis of the mitoQC mCherry signal. However, given the dramatic phenotypes observed with expressing this tether, much stronger than say loss of Pink1 or parkin (key mitophagy mediators), we think that it is unlikely that a decrease in mitophagy alone can explain the detrimental effects of increased tethering.

5. In Fig 6A,B the authors should include also the *pdzd8* KD to support their claim that the rescue of climbing defects correlates with an reduction of MERCs.

We thank the reviewer for this suggestion. We have performed this experiment and new Fig. 6A,B show that the increased number of MERCs in the AD model is indeed reduced by *pdzd8*-RNAi.

Moreover, it would be beneficial for their final conclusion, if the authors could show that increases mitophagy in the background of Ab42 expressing flies.

We thank the reviewer for this suggestion. In fact, we performed this analysis and found that while mitophagy increases upon A β ₄₂ expression, it was not further increased (or decreased) by *pdzd8*-RNAi. In light of this, we have adjusted our interpretation such that we are no longer able to conclude that the rescue is due to increase mitophagy. Instead, we further explored possible mechanisms and analysed the impact on mitochondrial calcium uptake where we saw a defect with A β ₄₂ expression and also a rescue by *pdzd8*-RNAi. We thank the reviewer for prompting us to explore the mitophagy aspect. In light of the new data, we feel this is now better represented and discussed in the manuscript.

Minor comments:

We thank the reviewer for their careful reading of the text and have corrected the issues below in the text.

1. Can the authors add to the figure legend of Fig 1F how the ER and Mitochondria were labeled?

We have added the constructs to the figure legend (full genotypes for all figures are given in Table S2).

2. Error bars should be added in the quantification of MERCs in Fig1C.

The MERCs are quantified in three brains per genotype but as there were variable numbers of sections suitable for imaging from each brain the total values are combined to give a single percentage.

3. A reference to Supplementary Fig S1D is missing in the main text.

This figure is referenced in line 148 (now rearranged as Figure S1E).

4. Can the authors label the individual genotypes in Fig S3C and 4F?

Figure labels and legends have been modified to clarify this.

5. Can the author specify which brain region they imaged in Fig 5C?

The regions imaged and quantified were chosen for their clear organelle morphology rather than targeting a specific brain region. All images were from the posterior protocerebrum and the methods and figure legends have been updated to note this.

6. Are the ATP levels normalized to ADP in Fig S3D? Can the authors specify in the figure and figure legend to what ATP was normalized?

Figure labels and legends have been modified to clarify the ATP levels are normalised to total protein quantification of the samples.

7. Please sort the supplementary figures in accordance to their reference order in the text.

We thank the reviewer for checking this. This has been carefully reviewed and placed in the appropriate order with the text.

Reviewer #2 (Significance (Required)):

The authors present here novel insights about the functional role of a new member of the MERCs, pdzd8, using RNAi mediated depletion and *Drosophila melanogaster* as a model system. As MERCs receive more attention especially in the context of their potential role in neurological diseases, the author's manuscript will be of high interest to the scientific community. The in vivo model combined with multiple different technical approaches add to the significance of the paper. There are some controls and additional experiments that are required to support the author's main claims and complete the functional characterization of pdzd8 (see major comments).

Field of expertise: neuroscience, fly genetics, neurodegeneration.

Reviewer #3

(Evidence, reproducibility and clarity (Required)):

This manuscript entitled "Decreasing pdzd8-mediated mitochondrial-ER contacts in neurons improves fitness by increasing mitophagy" by Hewitt and collaborators describes the role of the *Drosophila* ortholog of PDZD8 in ER-mitochondria contacts in neurons and the physiological consequence of pdzd8 loss. The authors show that ER-mitochondria contacts are reduced in fly neurons expressing a pdzd8-RNAi construct. Decreasing pdzd8 expression in neurons was

accompanied by a slowed age-associated decline in locomotor activity, and an increased lifespan. In presence of mitochondrial toxins, neurons deficient for *pdzd8* were protected. Finally, the authors showed that *pdzd8* silencing increased mitophagy in aged neurons, and protected against neurodegeneration in a model of Alzheimer's disease.

Major points:

1) There are important controls that are missing. RNAi expression often affects off-target genes which could unfortunately modify the observed phenotypes. The authors should verify that a) the phenotypes observed by RNAi-mediated *pdzd8* silencing can be rescued by the expression of an RNAi-insensitive *pdzd8* construct (the authors should verify the rescue of the most crucial phenotypes described in the manuscript); b) the RNAi-LacZ-line that they use as control in the paper does not behave differently from a WT line, which could be induced by an off-target effect of the RNAi-LacZ (again with the most crucial phenotypes).

While the *Drosophila* community is fortunate to have a plethora of readily available tools for interrogating the function of nearly all genes in the genome – tools which form the foundation of most work in *Drosophila* labs worldwide – the availability is not limitless. In this instance, the transgenic RNAi line generated as a resource for the community comprises a 500 bp hairpin, computed to be the most selective target for that gene. Being a 500 bp sequence it is unrealistic to be able to establish an RNAi-resistant variant that still faithfully functions as normal.

Nevertheless, although imperfect we show in Figure S4B that *pdzd8*-RNAi rescues the climbing defect produced by overexpressing *pdzd8*, providing evidence the construct is specifically acting on this sequence, and have noted this in the text.

Similarly, the availability of 'control' RNAi reagents is generous but still limited. This LacZ-RNAi line is one of a few well-established controls that has provided a cornerstone reference control for a wealth of studies. Nevertheless, we have revisited this and now provide experimental data that aged climbing of *nSyb>LacZ-RNAi* (in our view, the most reliable and crucial phenotype assessed here) is highly comparable to other well-established control genotypes including another control (luciferase) RNAi and an overexpression transgene (*mitoGFP*). These data are presented in new Fig. S3.

2) Did the author analyzed their EM data in a blinded-way to minimize subjective bias? This type of analysis is complicated by the manual annotation of ultrastructures, which is by nature subjective. For instance, this reviewer would have annotated the two mitochondria in the middle of Fig 1B, right as "Mitochondria with ER contact", as there is a membrane tube present at the interface of these two organelles.

Yes, the EM data were analysed blinded to the genotypes. This is noted in the methods section.

3) There is a controversy in the field on the role of PDZD8: some papers show its involvement in ER-mitochondria contacts, others in ER-lysosome contacts. The authors should discuss this point in more details. Moreover, the authors should localize the protein in *Drosophila* neurons; is the protein associated with mitochondria or endo/lysosomes?

We recognize that there is debate in the field over the localized role of PDZD8. However, since there is currently no antibody against the *Drosophila* protein and the sequence is sufficiently divergent such that antibodies against the mammalian protein do not recognize the fly protein (we have tried), we are not well-positioned to determine the localization of endogenous *Drosophila* *pdzd8*. Nevertheless, we have edited our discussion to reflect the differing views.

4) The authors should specify in more details how the different quantifications were performed.

For instance Fig 1G: how many samples were quantified (i.e. how many flies, and how many neurons); what is compared? Fields-of-view, neurons, flies...?

Further details have been added to the figure legends 1G (now H), 4G-I, 5 and Fig S2.

Minor point:

1) Could the authors show the SIM images Fig1 F together with the binarized images. These images have been added to Figure 1 and the legend and text updated accordingly.

2) It is surprising to see that data otherwise similar are represented with so many different types of graph (For instance Fig 5, bar graph, box-plot, violin plot). Why individual data points are not always present on the graphs?

We appreciate this point and always try to present the data in its most open yet accessible form. We have now unified our presentation modes, at least to be consistent within a particular analysis. Hence, we now present all data on MERCs, mitoQC, and measures of mitochondrial morphology and dynamics as box-and-whisker plots.

That said, while some datasets are amenable to certain presentation modes, others are not. For instance, our climbing assays produces discrete, discontinuous scores for individual flies in a cohort, and is not normally distributed (hence, the non-parametric statistical analyses).

Moreover, the scores are bounded between 0 and 1 and are typically performed on a large number of individual animals. So, trying to present the distribution of individual values in a grouping looks terrible whichever way we have tried. Over the years, we have determined that the most accessible yet fair way to present these data are as bar charts with mean and 95%CI.

3) The way that data are presented is sometimes odd: for instance, line 101, the authors wrote "To establish that MERCs were decreased...". This would imply that they knew the result before performing the experiment. And later, line 103 "Accordingly...".

These sentences have been rephrased "To determine whether MERCs were decreased.." and "These results showed the..."

Reviewer #3 (Significance (Required)):

This study about the role of pdzd8 is timely. The functional description of inter-organelle contacts is a hot topic in cell biology. There are several recent reports describing the identification of pdzd8 role in inter-organelle contact formation. This manuscript provides data on the role of pdzd8 in a whole organism and expands our understanding of this protein.

My expertise: inter-organelle contacts (human cells)

July 1, 2022

RE: Life Science Alliance Manuscript #LSA-2022-01531R

Dr. Alexander J Whitworth
University of Cambridge
MRC Mitochondrial Biology Unit
Hills Road
Cambridge CB2 0XY
United Kingdom

Dear Dr. Whitworth,

Thank you for submitting your Research Article entitled "Decreasing pdzd8-mediated mito-ER contacts improves organismal fitness and mitigates A β 42 toxicity". It is a pleasure to let you know that your manuscript is now accepted for publication in Life Science Alliance. Congratulations on this interesting work.

DISTRIBUTION OF MATERIALS:

Again, congratulations on a very nice paper. I hope you found the review process to be constructive and are pleased with how the manuscript was handled editorially. We look forward to future exciting submissions from your lab.

Sincerely,
